# BenthIQ: a Transformer-Based Benthic Classification Model for Coral Restoration

## Abstract

Coral reefs are vital for marine biodiversity, coastal protection, and supporting human livelihoods globally. However, they are increasingly threatened by mass bleaching events, pollution, and unsustainable practices with the advent of climate change. Monitoring the health of these ecosystems is crucial for effective restoration and management. Current methods for creating benthic composition maps often compromise between spatial coverage and resolution. In this paper, we introduce BenthIQ, a multi-label semantic segmentation network designed for high-precision classification of underwater substrates, including live coral, algae, rock, and sand. Although commonly deployed CNNs are limited in learning long-range semantic information, transformer-based models have recently achieved state-of-the-art performance in vision tasks such as object detection and image classification. We integrate the hierarchical Swin Transformer as the backbone of a U-shaped encoder-decoder architecture for local-global semantic feature learning. Using a real-world case study in French Polynesia, we demonstrate that our approach outperforms traditional CNN and attention-based models on pixel-wise classification of shallow reef imagery.

## 1 Introduction

Coral reefs play an integral role in preserving underwater biodiversity, providing habitats for roughly one-third of all marine species (McAllister, 1988). Shallow water reefs, in particular, protect coastal communities by reducing the impact of storms and erosion and provide crucial income for millions of people as a source of food and new medicine (Smith, 1978; Barbier et al., 2011). However, their corals are vulnerable to anthropogenic disturbances, such as pollution from urban runoff and agricultural fertilizer, non-sustainable harvesting, and coastal development activities (McAllister, 1988). Additionally, climate change-related stressors pose significant challenges for shallow coral reef environments, with the increasing frequency of mass bleaching events brought on by warmer water temperatures alone threatening the resiliency and survival of reefs worldwide (Hughes et al., 2017; Harrison et al., 2019).

Several coral research, conservation, and restoration organizations have been launched in the past decade in response. To monitor the aggregate effect of natural and human-induced stressors, determine resilient corals, and identify unhealthy areas in need of restoration, it is important to understand benthos distributions across the reef (Muller-Parker et al., 2015; Goldberg & Wilkinson, 2004). Thus, reef mapping missions are essential to supporting coral outplant efforts.

In situ benthos surveys for such mapping are limited to areas that are easily accessible by boat and are often inconsistent in terms of time, space, and scale. Over the past two decades, technological advances have rendered remote sensing a cost-effective and non-invasive solution to addressing these data gaps (Hedley et al., 2016). Imagery collected by satellites and airborne platforms can enable a higher frequency of consistent observations and effectively observe spatiotemporal changes in benthos distributions (Mumby et al., 2004; Phinn, 2011). With high-resolution drone data, it is possible to create precise benthic composition maps over entire reefs (Collin et al., 2018; Yasir Haya & Fujii, 2017; Saul & Purkis, 2015).

Deep learning methods for semantic segmentation can semi-automate the process of identifying and classifying underwater substrates in aerial imagery (Lirman et al., 2007; Kikuzawa et al., 2018; El-Khaled et al., 2022; Rich et al., 2022), potentially improving the efficiency and accuracy of segment-

ing complex objects such as coral colonies (Zhong et al., 2023; Zhang et al., 2022). Transformer-based models, in particular, have received attention in recent years following their success in the vision domain, and have increasingly been applied to segmentation tasks (Dosovitskiy et al., 2020; Thisanke et al., 2023; Li et al., 2023). In this work we (i) propose an encoder-decoder architecture with a transformer backbone for the semantic segmentation of high-resolution data, (ii) perform an ablation study to examine the impact of various model parameters, and (iii) explore the applications of this model in benthic composition mapping.

## 2 RELATED WORKS

In remote sensing applications, deep convolutional neural networks (CNNs) have been shown to outperform traditional machine learning methods (i.e. random forests, support vector machines, and conditional random fields) at feature extraction and object representations for image segmentation (Ma et al., 2019). UNet (Ronneberger et al., 2015), RefineNet (Lin et al., 2017), DFN (Yu et al., 2018), SegNet (Badrinarayanan et al., 2017), DeepLab v3+ (Yurtkulu et al., 2019), and SPGNet (Song et al., 2018) adopt a fully convolutional encoder–decoder structure to learn high-level semantic features and their spatial context. Specifically, the UNet and its variants, which have shown significant promise for medical image segmentation, consist of a symmetric encoder and decoder with skip connections (Ronneberger et al., 2015). The encoder uses downsampling for deep feature extraction with broad receptive fields, and the decoder upsamples these deep features to the input resolution to generate a mask with pixel-wise class predictions. The use of skip connections reduces spatial information loss during downsampling. Despite their powerful representation ability and efficiency, these CNN-based methods are inherently limited by local receptive fields and short-range context information (Fan et al., 2022; He et al., 2022; Thisanke et al., 2023; Li et al., 2023).

To capture long-range dependencies, Chen et al. (2017) proposed incorporating atrous spatial pyramid pooling (ASPP) with multiscale dilation rates to aggregate contextual information. The pyramid pooling module, introduced by Zhao et al. (2017), attempted to represent the feature map via multiple regions of different sizes. However, these context aggregation methods are still unable to sufficiently extract global contextual information.

Attention-based methods have been proven to be effective at obtaining global fields of view in semantic segmentation tasks (Niu et al., 2021). However, pixel-wise attention approaches use dense attention maps to measure the relationships between each pixel pair, posing computational and memory challenges. Moreover, attention-based methods are restricted to the perspective of space and channel, ignoring the class-specific information that is essential to semantic segmentation tasks. In the context of aerial data, feature representations of objects with the same category are different in complex scenes due to intra-class variation, context variation, and occlusion ambiguities. Therefore, dense pixel-wise attention tends to extract the wrong similarity relationship between pixels, leading to serious classification errors.

In response, researchers have begun to extend the success of transformers in the domain of natural language processing (NLP) to vision tasks (Vaswani et al., 2017; Carion et al., 2020). Dosovitskiy et al. (2010) proposes the vision transformer (ViT) for image recognition. In contrast to earlier attention mechanisms like dot-product attention, ViTs use structured patterns such as multi-head self-attention mechanisms which allow them to capture relationships between pixels at different positions in an image. When trained on large datasets, ViTs outperform CNN-based methods in object detection and image segmentation (Li et al., 2023). One such ViT, the hierarchical Swin Transformer, addresses the challenge of learning global contextual information using the shifted window attention mechanism and achieves state-of-the-art performance in vision tasks when used as a network backbone (Liu et al., 2021). Thus, we attempt to use the Swin Transformer block as the fundamental basis for a UNet-inspired architecture.

## 3 METHODS

Figure 1 depicts the proposed model architecture of BenthIQ, which follows the encoder-decoder structure of the UNet.

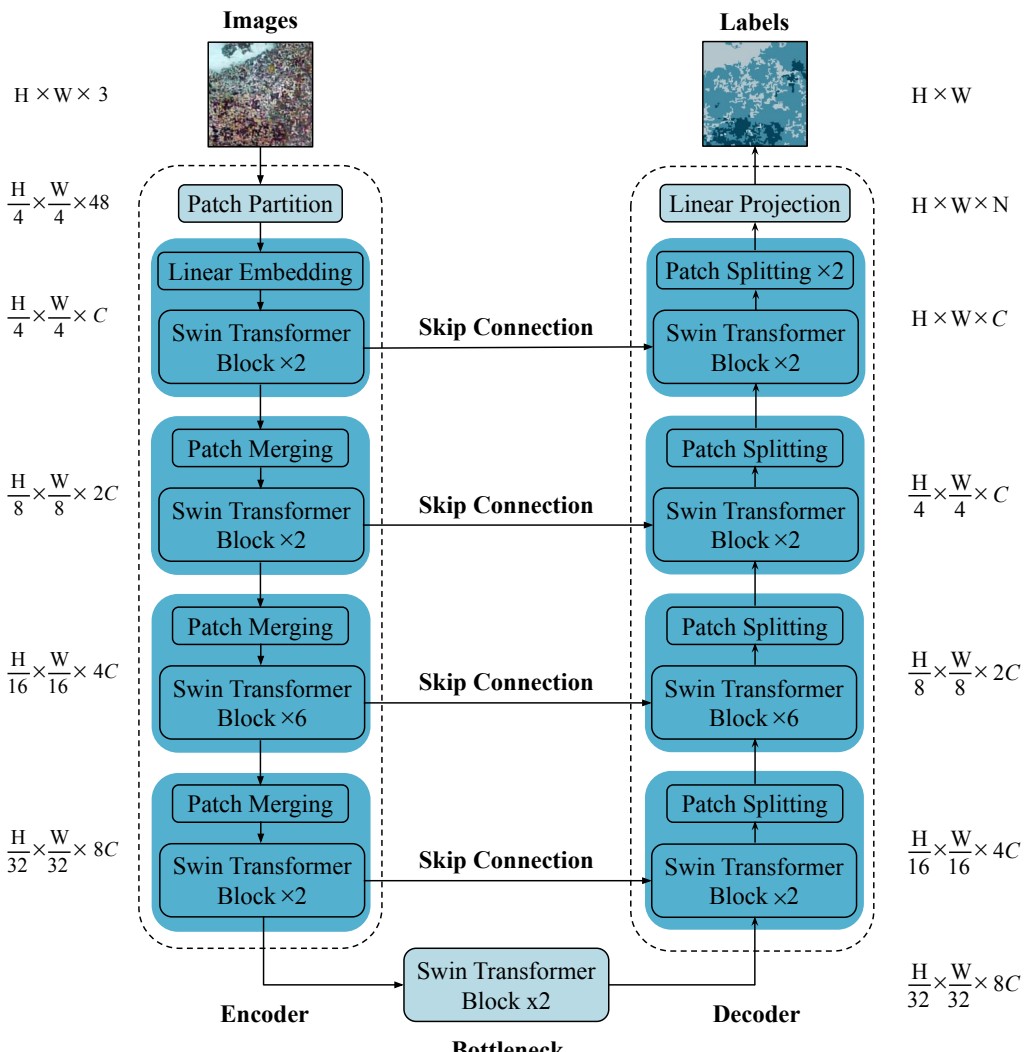

Figure 1: The U-shaped architecture of BenthIQ, which uses the Swin Transformer as a backbone. Here, $C$ is some arbitrary dimension, $N$ is the number of classes, and $H$ and $W$ represent the height and width of the input image, respectively.

## 3.1 SWIN TRANSFORMER BACKBONE

In the encoder and decoder, we refer to the implementation of the tiny Swin Transformer (referred to as Swin-T) from Liu et al. (2021). Every two consecutive transformer blocks consist of the window-based multi-head self-attention (W-MSA) module and the shifted window-based multi-head self-attention (SW-MSA) module to calculate the global attention, as shown in Figure 4 in the Appendix. A detailed W-MSA Block contains layer normalization (LN), a W-MSA module, and a multilayer perceptron (MLP) with GELU non-linearity. The LN normalizes the features to make the training process more stable, the W-MSA module calculates the attention relation between pixels, and the MLP contains a large number of learnable parameters to record the learned coefficients of W-MSA.

Instead of applying traditional MSA to calculate the attention relation in the whole $H \times W$ image, the Swin Transformer introduces W-MSA to calculate the attention relation in the $7 \times 7$ window size, greatly reducing the computational overhead. The SW-MSA addresses the challenges associated with reducing the receptive field to $7 \times 7$ when segmenting larger objects. By partitioning and merging the feature map between two transformer blocks and extending the local receptive field to

the global receptive field, the Swin Transformer efficiently captures spatial complexities and long-range dependencies.

## 3.2 ENCODER

As proposed in Liu et al. (2021), we begin by breaking down an input RGB image of shape $H \times W \times 3$ into distinct, non-overlapping patches using a patch partition module. Each patch is treated as a "token," with its feature represented as a combination of the raw pixel RGB values. In our implementation, we use a patch size of $4 \times 4$, resulting in a feature dimension of $4 \times 4 \times 3 = 48$ for each patch. A linear embedding layer then projects this raw-valued feature into an arbitrary dimension denoted as $C$.

To create a hierarchical representation, the number of tokens is reduced in the encoder through patch merging layers and sequences of Swin Transformer blocks. In the patch merging layer, we apply a linear layer to concatenate groups of four sub-patches. This results in $2\times$ downsampling and increases the feature dimension by $2\times$ (i.e. $\frac{H}{4} \times \frac{W}{4} \times C \to \frac{H}{8} \times \frac{W}{8} \times 2C \to \frac{H}{16} \times \frac{W}{16} \times 4C$, and so forth). The output is fed into sets of two Swin Transformer blocks (the W-MSA and SW-MSA modules), which maintain resolution and are responsible for feature representation learning. This process is repeated four times in the encoder.

## 3.3 DECODER

The bottleneck consists of two successive Swin Transformer blocks, which maintain the feature dimension and resolution, to learn the deep feature representation. In the decoder, we draw from the structure of the UNet to upsample the extracted deep features. We accomplish this using patch splitting layers and sequences of Swin Transformers with depths that correspond to the encoder. A linear layer is applied to the input to achieve $2\times$ upsampling, and we rearrange to reduce the feature dimension by a factor of $4\times$ (i.e. $\frac{H}{32} \times \frac{W}{32} \times 8C \to \frac{H}{16} \times \frac{W}{16} \times 4C \to \frac{H}{8} \times \frac{W}{8} \times 2C$, and so forth). In the final stage of the decoder, we repeat this patch splitting step twice to return the feature maps to the original input resolution. Finally, we apply a linear projection layer on the upsampled features to output the pixel-wise benthic labels in an $H \times W$ mask.

We use skip connections as proposed by (Ronneberger et al., 2015) to fuse the multi-scale features from the encoder with the upsampled features in the decoder. Shallow and deep features are concatenated to reduce the loss of spatial information caused by downsampling.

## 3.4 DATA

Our data was collected by The Nature Conservancy (TNC), and consists of an orthomosaic of the shallow reefs along the northern coast of Mo'orea, French Polynesia, with a ground sampling distance (GSD) of 1.1 cm per pixel. We partitioned this map into a grid, where each cell corresponds to a $224 \times 224$ RGB drone image. To generate the dataset, we randomly selected several $100 \times 100$ subgrids of the these images. This method is designed to maintain the integrity of complex data relationships on the local scale, while sampling images from different regions. Each image has a corresponding $224 \times 224$ mask with labels for sand, coral, algae, and rock. The resulting dataset contains 700,000 image-mask pairs, consisting of 48% sand, 23% coral, 12% algae, and 17% rock. Figure 5 in the Appendix visualizes sample data. The full classified dataset will be made publicly available by TNC upon completion.

## 3.5 TRAINING PARAMETERS

For all training and testing, we apply simple data augmentations, (e.g. random rotation and flipping) in addition to random color corrections (e.g. RGB shifts and brightness/contrast adjustments) to support model robustness under various lighting conditions. We use a 75-15-10 split for training, validation, and testing, with a batch size of 24. We use independent and identically distributed samples for our data splits to ensure that our model generalizes well to unseen data and captures spatial correlations in adjacent data patches. The training dataset was filtered to address the challenges associated with class imbalance, resulting in 312,774 image-mask pairs consisting of 21% sand, 31% coral, 20% algae, and 28% rock. All inputs are of size $224 \times 224$. For end-to-end remote sensing

Table 1: Ablation Study Results.

| Parameter | | mIOU | Sand | Coral | Algae | Rock |
|---|---|---|---|---|---|---|
| Input Size | 224×224 | 71.61 | 82.01 | 63.63 | 68.57 | 72.24 |
| | 512×512 | **74.51** | **84.70** | **65.39** | **70.51** | **77.43** |
| Upsampling | Bicubic Interpolation | 69.23 | 84.28 | 60.12 | 67.08 | 65.43 |
| | Max Unpooling | 70.03 | **84.84** | 62.41 | 67.93 | 64.93 |
| | Patch Splitting | **71.61** | 82.01 | **63.63** | **68.57** | **72.24** |
| Model Size | Swin-T (tiny) | 71.61 | 82.01 | **63.63** | 68.57 | 72.24 |
| | Swin-B (base) | **71.98** | **83.62** | 63.42 | **68.44** | **72.43** |

data segmentation, we initialize model parameters with Swin-T weights pre-trained on SEN12MS, a dataset of multi-spectral Sentinel-2 image patches and MODIS land cover maps (Schmitt et al., 2019). We minimize the Dice Loss (Li et al., 2019) for training, and we use the SGD optimizer with momentum 0.9 and weight decay 1e-4 to optimize our model for back propagation. Additional information on data augmentation, class balancing, and training is available in A.2 and A.3.

## 4 RESULTS

### 4.1 ABLATION STUDY

In order to explore the influence of different factors on the model performance, we conducted ablation studies on the Mo'orea reef dataset, the results of which are summarized in Table 1. Specifically, we discuss input sizes, upsampling methods, and model sizes below.

#### 4.1.1 ON THE INFLUENCE OF INPUT SIZE

We tested BenthIQ with the default input resolution of 224×224 and a higher-resolution setting of 512×512, fixing the patch size at 4. Increasing input size leads to a significant 4.04% improvement in mIOU. However, this improvement comes with a substantially higher computational cost. For the sake of computational efficiency, all experimental comparisons in this paper use the default resolution of 224×224 to showcase the effectiveness of BenthIQ.

#### 4.1.2 ON THE INFLUENCE OF UPSAMPLING

Our model uses a patch splitting layer in the decoder for upsampling and feature dimension enhancement. We evaluated the effectiveness of this new layer by comparing BenthIQ's performance using different methods like bicubic interpolation, max unpooling, and the patch splitting layer for 2× upsampling. Table 1 shows that BenthIQ with the patch splitting layer achieves superior segmentation accuracy.

#### 4.1.3 ON THE INFLUENCE OF MODEL SIZE

We examined the effect of network deepening on model performance. In particular, we experiment with the Swin-T ($C = 96$, layer numbers = {2, 2, 6, 2}) and Swin-B ($C = 128$, layer numbers = {2, 2, 6, 2}), where $C$ is the channel number of the hidden layers in the first stage (Liu et al., 2021). While the Swin-T is computationally more efficient and requires fewer resources for training and inference, the Swin-B is more capable of learning more intricate and detailed features from data as a larger model. From Table 1, it can be seen that the increase in model size results in minimal performance improvements (only by 0.5%), but significantly increases computational costs. Considering the trade-off between accuracy and speed, we adopt the Swin-T-based model to perform benthic classification.

### 4.2 BENTHIC CLASSIFICATION ANALYSIS

We evaluate model performance by computing the mean Intersection Over Union score (mIOU ↑) across the test set (Rahman & Wang, 2016). In Figure 2, we present sample model outputs

generated by BenthIQ. As anticipated, the error maps on the right demonstrate that the areas of highest uncertainty in model prediction are along the boundaries of substrates. Additionally, we observe that incorrect classifications are most often made around blurry or shadowed regions of the input image.

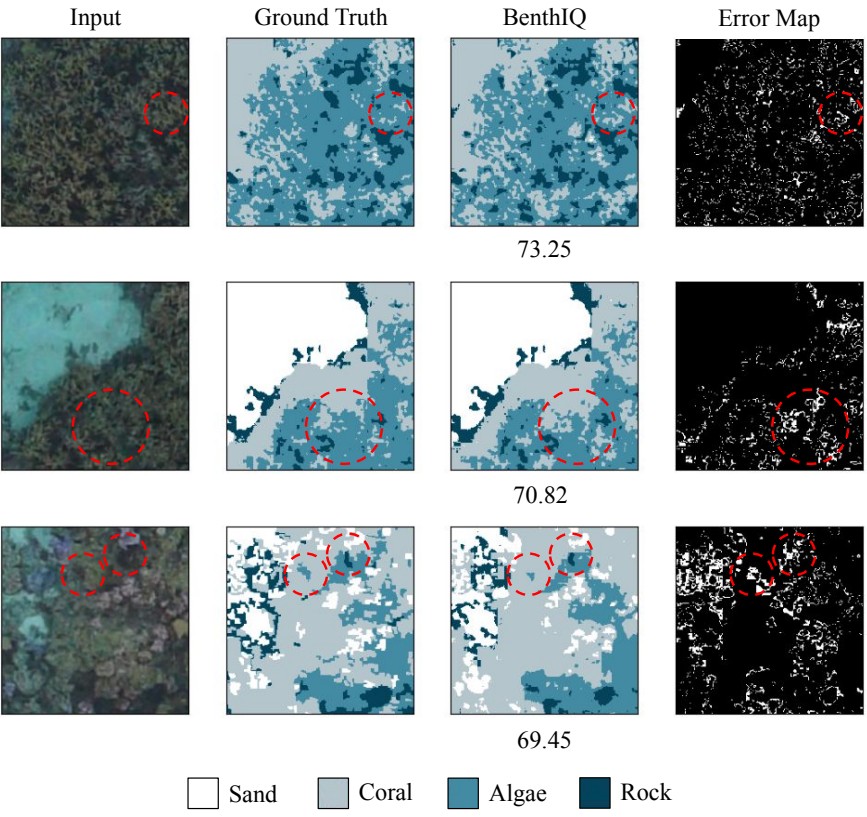

Figure 2: Visualization of BenthIQ performance on sample hand-selected inputs. From left to right: the input image, its ground truth mask, the corresponding BenthIQ output labeled with an mIOU score, and an error map with pixel-wise mismatches between the masks highlighted. Areas of interest are circled in red.

We compare the performance of our model against state-of-the-art CNN-, attention-, and ViT-based models: the ResNet50 UNet (Alsabhan et al., 2022; Ronneberger et al., 2015), ResNet50 Attn-UNet (Schlemper et al., 2019), ResNet50 ViT (Vaswani et al., 2017), and Efficient Transformer (Efficient T) (Xu et al., 2021). We fix the ResNet50 (R50) as the representative CNN backbone for standardized comparison (He et al., 2016). In Table 2, we report per-class IOU values, their mean, and border and interior accuracies. We define border regions to lie along the boundaries of substrates, with a width of two pixels. Over all classes, our model achieves the best performance, with an mIOU of 71.61. We note that our algorithm improves by 2.55% to 5.36% on the mIOU metric, and it achieves a 3.65% to 7.99% higher border prediction accuracy.

Across all models, sand is most accurately classified as it is lighter in color and much more easily distinguished from hard substrates (i.e. coral, algae, and rock). While rock is most often the dark and texture-less regions of the input aerial imagery, coral and algae are most spectrally similar and therefore hard to distinguish, resulting in lower IOU values for these classes across all models. Amongst pure CNN-based models, the UNet exhibits the lowest performance across all classes, while the Attn-UNet achieves high sand, coral, and rock accuracies. The latter method achieves similar results to BenthIQ, only outperforming our model by 0.15% in sand classification. Notably, it performs 8.56% worse in algae classification, suggesting that it often misclassifies small-scale algal growth as coral or rock.

Table 2: Model performance, averaged over the test set.

| Model | mIOU | Sand | Coral | Algae | Rock | Border | Interior |
|---|---|---|---|---|---|---|---|
| BenthIQ | **71.61** | 82.01 | **63.63** | **68.57** | **72.24** | **58.49** | **84.72** |
| Efficient T | 69.64 | 80.74 | 61.85 | 67.81 | 68.15 | 56.43 | 82.84 |
| R50 ViT | 67.97 | 81.38 | 58.39 | 62.45 | 69.65 | 55.88 | 80.05 |
| R50 Attn-UNet | 69.83 | **82.13** | 63.41 | 62.70 | 71.07 | 55.07 | 84.58 |
| R50 UNet | 66.32 | 80.87 | 61.50 | 61.01 | 61.89 | 53.41 | 79.23 |

The R50 ViT combines transformers with a CNN encoder without skip connections and produces inferior results to the pure CNN-based Attn-UNet. Additionally, while directly applying transformers for benthic classification yields reasonable results (69.64 mIOU for the Efficient T), this approach results in a similar performance to the Attn-UNet. This is likely due to the fact that while pure transformer models are capable of learning high-level semantics, they lack low-level cues for classification on finer spatial scales.

Our model builds upon these pure transformer and attention-based approaches using a UNet structure with skip connections. With these improvements, BenthIQ seems able to learn both high-level semantic features and the low-level details of small and irregularly shaped hard substrates and achieves the best mIOU performance, particularly in border regions. Unlike traditional CNN-backbones which have limited receptive fields, the Swin Transformer employs a hierarchical structure that enables it to process information across the entire input. This renders it particularly effective as the basic unit for benthic classification, which requires both local and global contextual understanding. BenthIQ's higher accuracy in classifying hard substrates (algae, in particular) indicates that it may be better at learning complex data relationships locally and generalizing to other regions of the reef with varying benthic compositions.

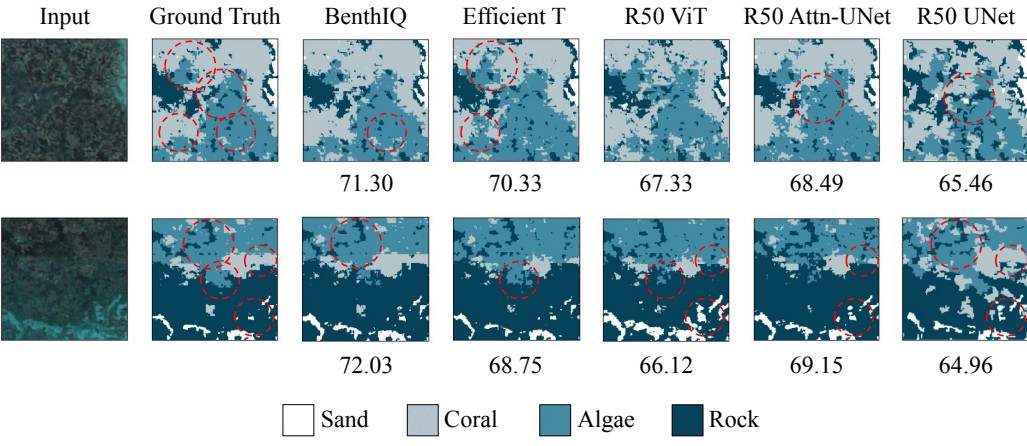

Figure 3: Qualitative comparison of different models, based on hand-selected inputs with high coral, algae and sand cover. From left to right: the input image, its ground truth mask, and the outputs of BenthIQ, Efficient T, ResNet50 ViT, ResNet50 Attn-UNet, ResNet50 UNet. Areas of interest in model outputs are circled in red. In these examples, our method seems to preserve information on finer spatial scales.

In Figure 3, we provide a qualitative comparison of model performance. We observe that the pure CNN-based methods (the UNet and Attn-UNet) often over-segment or under-segment substrates, likely due to the locality of the convolution operation. This is exemplified in the second row of the figure, where the Attn-UNet undersegments the rock and oversegments the coral. The UNet generates coarse edge predictions for all classes and overclassifies both sample inputs. Amongst the ViT-inspired models, we observe that the R50 ViT undersegments rock and sand and misclassifies coral as algae in the second example. While the Efficient T achieves precise edge predictions as shown in the second row, it often overestimates rock and algal cover in the shadowed regions of

the input image, as seen in the first row. We attribute the relative success in edge prediction to the Edge Enhancement Loss used during Efficient T training (Xu et al., 2021). Overall, BenthIQ most accurately classifies hard substrates, with fewer misclassifications between coral and algae, which are the most challenging to differentiate.

## 5 DISCUSSION

Our analysis demonstrates that BenthIQ achieves state-of-the-art performance in pixel-wise benthic classification. Our model improves upon traditional CNN-based approaches with the Swin Transformer, which uses the shifted window attention approach to identify finer features and learn long-range semantic information. In the context of benthic composition mapping, this is particularly useful in identifying irregular and small-scale substrates. Unlike existing transformer-based models, our UNet-inspired architecture with skip connections maintains local spatial details, making it effective at capturing high-frequency details even in the presence of downsampling in the encoder. BenthIQ outperforms other models in classifying hard substrates, coral and algae in particular. Its improvement in predictions along the boundaries of substrates further demonstrates its ability to accurately achieve classification on fine spatial scales.

BenthIQ's ability to accurately and precisely classify benthic composition is a crucial asset for reef restoration efforts. We have shown that the model's capacity to learn semantic information on the local and global scale is particularly invaluable in segmenting irregular algal growth and complex reef and rock structures. In improving upon the precision and border prediction accuracy of existing semantic segmentation methods, we can better isolate potential mother colonies to extract coral fragments from and identify rocks or dead substrates that are suitable in size and shape for hosting these fragments in the outplant process. Our accuracy gains are also essential to benthic composition calculations for planning and monitoring restoration. Specifically, identifying areas with diminished live coral cover and high algal concentration can help prioritize outplanting in at-risk reefs. Additionally, in comparing composition maps over broad temporal scales, it is possible to bettter understand the impact of and respond to environmental stressors such as ocean warming events, pollution, invasive species, or disease.

Future research in this domain may center around incorporating an Edge Enhancement Loss, as used in the Efficient T, into the training objective. This may yield additional performance boosts in border predictions, which is especially difficult in overwater imagery where algae, coral, and rock overlap and form complex outlines.

While pre-training on SEN12MS satellite data was more domain-specific to our remote sensing application than another large image segmentation dataset such as ImageNet (Deng et al., 2009), our model may achieve minor performance boosts when pre-trained on overwater or underwater reef imagery instead. Using reef data may help the model adapt to common underwater features and conditions, such as lighting, water clarity, and complex reef structures, which could improve its performance when applied to aerial imagery. Additionally, augmenting the training dataset with benthic composition maps from different geographic regions with a variety of coral and algae species may yield a more generalizable and robust model. Although restoration scientists are primarily interested in a coarse ontology (identifying areas of generalized coral and algal cover), we hope to evaluate our model performance on classifying specific species of coral and algae on the biological family level in future iterations of this work.

## 6 CONCLUSION

In this work, we introduced a novel encoder-decoder architecture with a ViT backbone for the semantic segmentation of aerial reef imagery. Using Swin Transformer blocks for learning short- and long-range semantic information and feature representation, our proposed model achieves state-of-the-art performance in pixel-wise classification of sand, coral, algae, and rock in data sampled from the shallow reefs in French Polynesia. BenthIQ's performance in this study underscores its potential for enhancing coral reef monitoring and restoration efforts, and our methodology may be extended to high precision classification tasks in other domains.

ETHICS STATEMENT

This work aims to introduce a non-invasive, efficient, and accurate method for benthic composition mapping. Since the focus of this study is on Mo'orea, French Polynesia, our results may not be generalizable to other geographic regions. However, we provide a replicable codebase that will enable transition of our results to other national wind and solar datasets. Our work spans ecological monitoring, conservation, and restoration, and we are mindful of ethical implications. We have no conflicts of interest and adhere to all legal requirements, committed to responsible innovation and stakeholder engagement.

REPRODUCIBILITY STATEMENT

We prioritize transparency, research integrity, and reproducibility by sharing our code and model weights. The data for this analysis was collected and processed by TNC, and full dataset will be made publicly available upon completion. We share and our code for generating visualizations of BenthIQ outputs. Our code will be published and open source upon acceptance. Refer to 3.4, 3.5, A.2, and A.3 for more information on our data processing and training configurations.

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

## A  APPENDIX

### A.1  SUPPLEMENTARY FIGURES

Below, we include supplementary figures referred to in the paper.

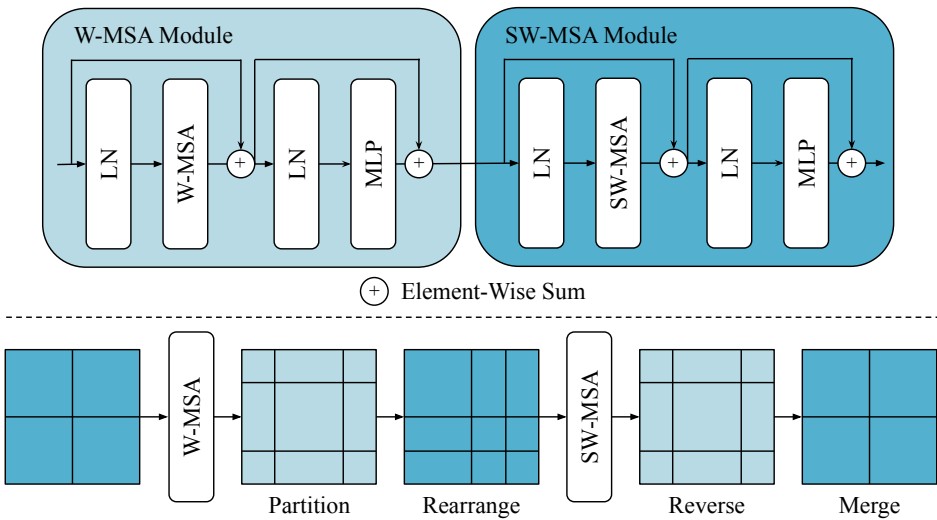

Figure 4: Swin Transformer blocks (top) and the shifted window approach (bottom).

### A.2  DATA PROCESSING

For data augmentation, we first randomly rotate the input image and label by 90, 180, or 270 degrees and also perform horizontal or vertical flipping with a 50% probability. Then, we randomly rotate the input image and label by an angle between -20 and 20 degrees. This helps the model learn variations in object orientation and appearance. With a probability of 50%, we apply random shifts to the red, green, and blue channels of the image with a tolerance of 20 to account for various lighting and water surface conditions. Lastly, we apply random adjustments to the brightness and contrast of the image with a probability of 50%. First, we center the pixel values of the image around 0 and adjust by a multiplicative contrast factor (randomly chosen between -0.3 and 0.3). Then, we recenter the values to 128 and adjust by an additive brightness factor (randomly chosen between -0.3 and 0.3).

In the original dataset, relative algal concentrations are low. To enforce a fair representation of all classes during training, we filter the dataset using stratified sampling. Specifically, we consider randomized mini-batches of size 24, and ensure that each mini-batch contains a proportional representation of each class. Specifically, we include mini-batches which include image-mask pairs class abundance between 20% to 40% for all classes.

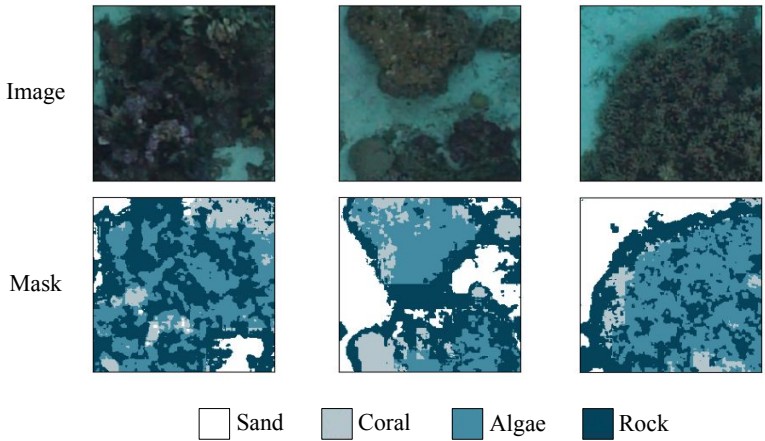

Figure 5: Sample images (top), and their corresponding masks (bottom) from the Mo'orea reef dataset.

## A.3 TRAINING PARAMETERS, EXTENDED

We train for 500 epochs on an NVIDIA Tesla T4 GPU, for a total training time of approximately 7 hours. We fix the random seed to 1234, and enforce deterministic CUDA neural network operations. We share a sample dataset of 5 image-mask pairs randomly chosen from our test data, the pretrained Swin-T weights, our BenthIQ weights. Our code is available for generating visualizations of our model outputs.

