# OpenReview forum: "BenthIQ: a Transformer-Based Benthic Classification Model for Coral Restoration"
_ICLR.cc/2024/Conference — ICLR 2024 Conference Withdrawn Submission_

### Official Review · Reviewer_cnzW · 2023-10-30

**Soundness:** 3 good
**Presentation:** 3 good
**Contribution:** 2 fair
**Rating:** 5
**Confidence:** 4

**Summary:**

This paper shows a semantic segmentation model applied to coral aerial photo segmentation. The model is based on a U-net shaped architecture with Swin transformer as backbone. The authors releases a new dataset, on which they made a comparison between state-of-the-art semantic segmentations models and their model.

**Strengths:**

The results are surpassing other models, yet not necessarily largely.
The paper is well-written, it proposes an interesting application and a new dataset.

**Weaknesses:**

The model that was proposed as 'new' is actually a well-known model, so there is no methodological novelty. Unless the authors prove me wrong on the following points (in particular the first one), I don't think it is worth publishing in ICLR.
1) By reading the paper, it looks as if the model proposed was constructed by the authors, while it is an exact copy of the 'Swin-Unet: Unet-like Pure Transformer for Medical Image Segmentation', Cao et al. 2022 (cited 1300 times). This paper is not even cited. Please comment and explain.
2) It would be important to also compare your model on existing databases and thus existing benchmarks. Is there a reason why you did not?
3) The release of data is not fully clear. It is said that it will be released at the paper acceptance, but on another part of the paper it is said that it will be released 'by TNC upon completion of the database': will that be at the paper acceptance?

**Questions:**

cf. section weakenesses.

Q1) As I understand, the dataset is new release coming with the paper. From my perspective, it can have an
interesting value, but you would need to better explain the advantage of this database compared to existing ones, and also how was the labelling performed, and with which uncertainty?


Typos:
- of the these images
- We share and our code

---

### Official Review · Reviewer_G8Fd · 2023-10-31

**Soundness:** 2 fair
**Presentation:** 3 good
**Contribution:** 1 poor
**Rating:** 3
**Confidence:** 5

**Summary:**

This work addresses the semantic segmentation on benthic images. The designed BenthIQ is a U-shaped neural network with Swin Transformer and skip connections for the semantic segmentation of benthic images. The collected dataset contains benthic images labeled with four pixel-level categories of sand, coral, algae, and rock. The authors analyzed the performance of BenthIQ on the collected dataset in terms of ablation study and benthic classification.

**Strengths:**

- It is meaningful to collect real-world benthic images with semantic labels for the study of coral reef restoration and protection.

**Weaknesses:**

- The novelty of this work is limited. The designed method, namely BenthIQ, is a general architecture that is commonly used in many fields, thus it has very limited contributions about the network. Although the collected dataset is helpful, the authors should provide more details about this dataset as well as why and how to build this dataset for the study of coral reef restoration and protection.
- The experiments are insufficient. The analysis is mainly about how some parameters influence the performance, and the comparison doesn't consider many cutting-edge semantic segmentation methods. Actually, it might be meaningful to dive deep into the real-world applications of this study for coral reef restoration and protection, and it might be better if the authors consider to firstly build a benchmark with state-of-the-art semantic segmentation methods and then design a new method to improve the performance further for meeting the requirements of real-world applications on coral reef restoration and protection.

**Questions:**

This work is still preliminary on semantic segmentation of benthic images, and the authors are suggested to dive deep into the topic of coral reef restoration and protection with cutting-edge artificial intelligence techniques.

---

### Official Review · Reviewer_pc9S · 2023-11-01

**Soundness:** 3 good
**Presentation:** 3 good
**Contribution:** 3 good
**Rating:** 6
**Confidence:** 3

**Summary:**

This paper addresses the crucial task of monitoring coral reef health, highlighting the current trade-off between spatial coverage and resolution in existing methods. The authors introduce BenthIQ, a novel multi-label semantic segmentation network that leverages the hierarchical Swin Transformer within a U-shaped encoder-decoder architecture for underwater substrate classification. This fusion allows the model to effectively learn both local and global semantic features. Through a case study in French Polynesia, they convincingly show that BenthIQ surpasses both traditional CNN-based and attention-driven models in pixel-wise classification of shallow reef imagery.

**Strengths:**

1. The work has great potential to contribute to oceanography, leveraging ViT.
2. Good writeup style, in terms of grammar and readability.

**Weaknesses:**

1. Research questions are not specified. It is better to specify it in bullet points, like you did in mentioning contributions.
2. It is not mentioned how the parameters and/or hyperparameters of different architectures are dealt with.
3. No limitation is mentioned here.

**Questions:**

1. Table 1,2: The results are not clear to me. I mean I understand the mIOU part, but about Sand, Coral, Algae, Rock, what are the numbers of? Is it accuracy? Or which metric exactly? This is not identifiable at a first glance. Please indicate that and add that in the caption of Table 1. This will be beneficial, especially for new researchers in this field.
2. The architectures compared in this work are diverse enough and have a lot of different hyperparameters, some which are commonly shared and others which are not. Could you please add more discussions about what steps you took to ensure that comparisons were fair and not biased by something which you can actively avoid?
3. Which part of your network specializes in this Coral Detection task if it is just a Swin Transformer?
4. Future Work should not be in the discussion section. You can update the section “Conclusion” to “Conclusion and Future Work”.
5. You have written that you implemented “Swin Transformer” as the backbone in abstract, but in the conclusion, it is written as “ViT” as the backbone. It is better to specify that you used a variation of ViT in the conclusion.